


# The Floodwater Depth Estimation Tool (FwDET v2.0) for Improved Remote Sensing Analysis of Coastal Flooding

Sagy Cohen[1], Austin Raney[1], Dinuke Munasinghe[1], Derek Loftis[2], Andrew Molthan[3], Jordan Bell[3], Laura Rogers[4], John Galantowicz[5], G. Robert Brakenridge[6], Albert J. Kettner[6], Yu-Fen Huang[7], Yin-Phan Tsang[7]

[1]Department of Geography, Univeristy of Alabama, Tuscaloosa, 35487, USA
[2]Virginia Institute of Marine Science, College of William and Mary, Gloucester Pt, 23062, USA
[3]SPoRT Center, NASA Marshall Space Flight Center, Huntsville, 35808, USA
[4]Disasters Program, NASA Langley Research Center, Hampton, 23666, USA
[5]Atmospheric and Environmental Research, Inc. (AER), Lexington, 02421, USA
[6]Dartmouth Flood Observatory, University of Colorado, Boulder, 80309, USA
[7]Natural Resources and Environmental Management, University of Hawaii, Manoa, 96822, USA

*Correspondence to*: Sagy Cohen (sagy.cohen@ua.edu)

**Abstract.** Remote sensing analysis is routinely used to map flooding extent either retrospectively or in near-real-time. For flood emergency response, remote sensing-based flood mapping is highly valuable as it can offer continued observational information about the flood extent over large geographical domains. Information about the floodwater depth across the inundated domain is important for damage assessment, rescue, and to prioritize relief resource allocation, but cannot be readily estimated from remote sensing analysis. The Floodwater Depth Estimation Tool (FwDET) was developed to augment remote sensing analysis by calculating water depth based solely on an inundation map with an associated Digital Elevation Model (DEM). The tool was shown to be accurate and was used in flood response activations by the Global Flood Partnership. Here we present a new version of the tool, FwDET v2.0, which enables water depth estimation for coastal flooding. FwDET v2.0 features a new flood boundary identification scheme which accounts for the lack of confinement of coastal flood domains at the shoreline. A new algorithm is used to calculate the local floodwater elevation for each cell, which improves the tool's runtime by a factor 15 and alleviates inaccurate local boundary assignment across permanent water bodies. FwDET v2.0 is evaluated against physically-based hydrodynamic simulations in both riverine and coastal case studies. The results show good correspondence, with an average difference of 0.18 m and 0.31 m for the coastal (using a 1-m DEM) and riverine (using a 10-m DEM) case studies respectively. A FwDET v2.0 application of using remote sensing derived flood maps is presented for three case studies. These case studies showcase FwDET v2.0 ability to efficiently provide a synoptic assessment of floodwater. Limitations include challenges in obtaining high-resolution DEMs and increases in uncertainty when applied for highly fragmented flood inundation domains.

## 1 Introduction

Flooding is the most destructive natural disaster on Earth. About 100,000 people lost their lives due to floods in the last decade of the 20th century (Higgins et al., 2014). The highest loss proportion of the global insured catastrophes in 2017 (144 billion USD) came from Hurricanes Harvey, Irma, and Maria, resulting in combined insured losses of $92 billion (Swiss Re, 2018). Of the global disasters between 1994 and 2013, 43% were floods, affecting approximately 2.5 billion people (CRED, 2015). Coastal regions are particularly susceptible to flooding due to their low gradient terrain and exposure to storm surges and tsunamis (Li et al., 2018). Sea level rise, coupled with land subsidence and rapid urbanization, has led to increased flood risk in many coastal regions worldwide (Tessler et al., 2015). Monitoring, analyzing, and forecasting floods are commonly based on numerical models of hydrodynamic and meteorological processes, *in situ* gaging, and remote sensing analysis. The application of these tools and techniques for the unique topography of coastal flooding events is often problematic due to the low topographic gradients, greater diversity in flooding mechanisms, and complex riverine-coastal interactions. Hydrodynamic models typically rely on terrain data to simulate flood fluid dynamics (e.g. the GSSHA and LISFLOOD-FP models). Low variability in coastal topography heightens

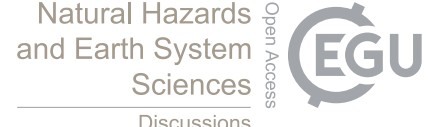



the requirements of high-resolution elevation data (e.g. LiDAR DEM), which, if available, can increase runtime and introduce numerical instabilities. Unlike confined floodplains, *in situ* gaging (e.g. tide gaging) cannot be easily translated into flood extent and severity estimates. This challenge is a product of coastal terrain and floodwater origin complexity (i.e. coastal, river and pluvial water accumulation).

5        Remote sensing-based analysis of flooding, which is largely agnostic with respect to flooding mechanisms and sources, can be used to rapidly generate flood extent maps in near-real-time. These analyses often apply standard algorithms and tools, and for most first-order remote sensing approaches there is no need for supplementary data. Remote sensing has substantial advantages over modeling approaches, especially for emergency response and large-scale analyses, and particularly in coastal regions where accurate flood extent simulations can be challenging (Gallien, 2016). However, the disadvantages of remote sensing approaches

include limitations in imagery availability and acquisition time, coarseness of resolution, cloud cover (for optical sensors), nonlinearities in signal reflectance (particularly for radar sensors), and view obstruction by vegetation, topography, buildings, and their shadows. Remote sensing also cannot be readily used to map water depths.

        Timely information about floodwater depth is important for directing rescue and relief resources and determining road closures and accessibility. Once available, flood depth information can also be used for post-event analysis of property damage

and flood-risk assessment (Islam and Sadu, 2001; Nadal et al., 2009; Nguyen et al., 2016). Several approaches for quantifying floodwater depth using remote sensing-based flood maps have been proposed. Nguyen et al. (2016) combine a flood extent map with hydrodynamic simulations. While accurate, this approach is both data- and computation-expensive, thus hindering its usability for data-scarce, near-real-time, and large-scale applications. Schumann et al. (2007) develop a floodwater depth calculation model based on high-resolution flood extent and DEM layers. Their model uses regression analysis to interpolate between Hydrologic

Engineering Center's River Analysis System (HEC-RAS) cross sections along the flooded domain. Cohen et al. (2018a) use a somewhat similar concept but instead of cross sections, their Floodwater Depth Estimation Tool (FwDET) identifies the floodwater elevation for each cell within the flooded domain based on its nearest flood-boundary grid-cell (described in more detail below). As a result, FwDET removes the need for specific data while retaining its usability with complex and fragmented flood extent maps from any source and resolution (i.e., sensor and platform independent).

Since its development in 2017, FwDET has been used in support of emergency response as part of activations of the Global Flood Partnership (GFP; https://gfp.jrc.ec.europa.eu; Alfieri et al., 2018), including the 2017 and 2018 U.S. Hurricane Seasons (Cohen et al., 2018b) and 2018 Philippine and Nigeria flooding. It was also been recently used for disaster resilience research (Loftis et al., 2018, Rogers et al., 2018, NASA CAIR, 2018). Findings from these activities are described in Cohen et al., 2018b, including the previously described challenges in coastal flood analysis. The need for fine resolution terrain data (to account

for low gradients) mandate considerable improvements in FwDET computational efficiency to reduce run-time. Flood inundation polygons, used in FwDET to identify flooded domain boundary locations (grid-cells), inevitably include those on the shoreline or ocean water (where elevation is equal or below mean sea level), and both introduce erroneous water depth calculations in nearby grid-cells. Furthermore, complex shorelines (e.g., small bays, inlets, barrier islands) can result in nearest flood-boundary cells erroneously located across a waterbody. In this paper, we describe and evaluate version 2.0 of FwDET, which was developed to

alleviate these issues. FwDET v2.0 was developed as part of the NASA Applied Sciences Mid-Atlantic Communities and Areas at Intensive Risk (CAIR) demonstration project (Rogers et al., 2018) and its application within the project is described here. While developed primarily to address coastal issues, FwDET v2.0 retains its applicability to estimating riverine floodwater depth. The use of FwDET v2.0 for riverine flooding is also analyzed herein.

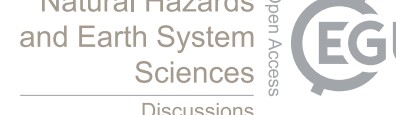

## 2 Methodology

### 2.1 FwDET v2.0

FwDET calculates water depth by deducting local floodwater elevation (above mean sea level (amsl)) from the topographic elevation at each grid-cell within the flooded domain. The flooded domain is provided as a GIS polygon layer to FwDET, making

the tool agnostic to the source and method used to derive the inundation extent. Elevation of each grid-cell and the floodwater is derived from a Digital Elevation Model (DEM). While any DEM can be used, its horizontal and vertical resolutions can have a major impact on the tool's accuracy. This is discussed in more detail below as well as in Cohen et al. (2018a and 2018b). The core of the FwDET algorithm is the identification of local floodwater elevation. FwDET water depth calculation follows this following procedure (described and illustrated in detail in Cohen et al., 2018a): (1) conversion of the inundation polygon to a line layer, (2)

creation of a raster layer from the line layer that has the same grid-cell size and alignment as the DEM, (3) extraction of the DEM value (elevation) for these grid-cells (referred to as boundary grid-cells), (4) allocation of the local floodwater elevation for each grid-cell within the flooded domain from its nearest boundary grid-cell, and (5) floodwater depth calculation by deducting local floodwater elevation from topographic elevation at each grid-cell within the flooded domain.

For flooding within a river floodplain, associating the appropriate boundary grid-cell is relatively straightforward as

illustrated in Figure 1 (top) with a cross-section. In non-continuous flood domains (e.g. in floodplains of braided rivers), isolated areas of non-flooded land can, and quite often, exist. Non-flooded isolated areas can be real or represent an error in the remote sensing analysis due to, for example, undetected flooding under dense vegetation. FwDET identifies the cells around these areas as boundary grid-cells, which, if these are real non-flooded (elevated) areas, is expected to improve the water depth calculations as it provides more localized floodwater elevation data. In coastal floods, the inundation polygon boundary at the coastline or

ocean waters cannot be used as boundary grid-cells as the DEM-extracted elevation will not represent the floodwater depth as illustrated in Figure 1 (bottom). These boundary grid-cells should, therefore, be excluded from the analysis. In FwDET v2.0 this is done by removing all boundary grid-cells that have or are immediately adjacent to grid-cells that have an elevation equal to or less than zero. The inclusion of adjacent cells in this conditioning is done as coastal inundation polygons will often end at the coastline and the conversion to a raster will often result in boundary grid-cell immediately inland of the coastline, resulting in

elevation (depending on the DEM resolution) that can be slightly greater than zero.

The first version of FwDET (v1.0) was implemented using a Python script which utilizes ArcGIS tools (ArcPy library) for its core data analysis (available at https://sdml.ua.edu/models/ and https://csdms.colorado.edu/wiki/Model:FwDET). Floodwater elevation of the nearest boundary grid-cell is allocated in FwDET v1.0 by iterating over increasing neighborhood sizes of the ArcGIS 'Focal Statistics' tool (ESRI, 2019a). The iteration includes a condition to ensure newly allocated flooded grid-cells

receive elevation values from their closest computed neighbor (i.e. nearest boundary grid-cell). This approach has three disadvantages: (1) it requires running the 'Focal Statistics' tool multiple times, reducing FwDET computational efficiency; (2) the size of the largest neighborhood needed to cover the entire flooded domain varies depending on the domain size and the DEM resolution, requiring an *a priori* estimation of the number of iterations, often resulting in the need to re-run the tool; and (3) it ignores permanent water features (rivers, inlets), and thus can erroneously assigns boundary grid-cell elevations to flooded grid-

cells on the opposite bank because their Euclidian distance is shorter than to the boundary grid-cells on their side of the waterbody.

In FwDET v2.0, allocation of the nearest boundary grid-cell elevation is done with the ArcGIS 'Cost Allocation' tool (ESRI, 2019b). 'Cost Allocation' changes the way in which nearest boundary grid-cells are allocated to a non-iterative approach. This drastically reduces the run time, as the tool uses one linear process to allocate the value of the input raster's (boundary elevation raster) nearest grid-cell for all cells within the output domain. The tool's 'cost' input raster is used in FwDET v2.0 to

prevent boundary grid-cell elevation allocation over permanent water by assigning such grid-cells with high-cost value. The cost



raster is calculated by assigning all grid-cells with elevation equal to or less than zero a value of 1000 and all other grid-cells a value of 1. For inland water bodies (e.g., rivers; where permanent water bodies have greater than zero elevation), a cost raster can be calculated from a land-cover map and used as input to FwDET v2.0. The 'Cost Allocation' tool only accepts integers, consequently creating a vertical elevation data resolution of 1 elevation unit (e.g. meter), which was the main reason this tool was not used in previous versions of FwDET. In FwDET v2.0, a float-integer-float conversion is employed to maintain the DEM vertical resolution.

FwDET v2.0 is also available as a Python script and as an ArcGIS Script Tool (see Conclusions section). A QGIS Python script and tool was developed to eliminate dependency on ArcGIS licensing. The QGIS script runtime is shorter than the ArcPy-dependent script but does not yet include a cost raster input and therefore does not solve the above-indicated issue of allocation across permanent water.

## 2.2 Evaluation

FwDET v2.0 water depth estimations are evaluated here against calibrated hydrodynamics simulations. A similar approach was used in Cohen et al., (2018a) to evaluate FwDET v1.0. The flood extent used as input for FwDET v2.0 is derived from the modeled water depth rasters. This allows for the evaluation of FwDET v2.0 floodwater depth calculation approach without introducing potential biases by using different inundation extent data. The same DEM were used for the models simulations and FwDET v2.0 calculation in each case study. The (calibrated) model-simulated water depth is assumed here to be the true water depth. Biases in this analysis are therefore solely a function of FwDET v2.0 algorithm results relative to the model's fluid dynamic simulation. Two evaluation case studies are presented:

1.  Portsmouth and Norfolk (Virginia, U.S.A.) – coastal flooding in a relatively flat tidewater urban environment during Hurricane Irene in 2011. A 1-m topobathymetric LiDAR DEM, developed by Danielson et al. (2016), provided elevation inputs to two separate hydrodynamic models. These models operate at two different scales, large scale (20 m – 10 km) to street-level scale (1 m – 10 m). SCHISM (Semi-implicit Cross-scale Hydroscience Integrated System Model) covered half of the North Atlantic Ocean to the coastal zone with individual grid cells ranging from 20 m – 10 km. The UnTRIM[3] hydrodynamic model (Casulli, 2019), resolved an area the size of the Portsmouth and Norfolk, respectively, and their adjacent waterway systems at the street-level scale (1 - 10 m resolution grid, Wang et al., 2014). SCHISM is an open-source hydrodynamic model (Zhang et al., 2016), which was provided tidal harmonic inputs at the open boundary, and atmospheric inputs for Hurricane Irene from two different atmospheric model simulations. These atmospheric models were the Weather Research and Forecasting (WRF) model using nested 9, 3, and 1 km resolution grids, with hourly time intervals, and the European Center for Medium Range Weather Forecasting (ECMWF) model at 12 km resolution using 3-hour time intervals. Water level outputs from SCHISM were used as Dirichlet open boundary inputs for the street-level model to drive inundation into urban environments and highlight vulnerable infrastructure impacted during the storm. These urban structures were extracted from LiDAR as in Loftis and Taylor (2018) and directly embedded in the model to account for volume displacement within the structure's surface area and form drag as the storm surge flows around each building within the urban environment (Loftis et al., 2014, 2016). The street-level model subsequently generated hourly inundation outputs for floodwater depths in meters, which compared favorably with water level sensors and high water marks as noted in Loftis et al. (2018). A more complete description of this case study is available in Loftis et al., (2018), and Rogers et al., (2018). In this paper, the maximum water depth output was used along with the 1m DEM for the FwDET v2.0 calculation.

2.  Brazos River (Texas, U.S.A.) – riverine flooding during the May 2016 flood event. The same event was used to evaluate FwDET v1.0 and is used here to compare the two FwDET versions for riverine flooding. The iRIC-FaSTMECH

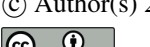



hydrodynamics model (Nelson et al., 2016; https://www.i-ric.org) was used to simulate the flood. The iRIC-FaSTMECH simulates water velocity and water surface elevation using gaged discharge input at the reach's upstream location and stage at its downstream outlet. Manning's roughness parameter was calibrated, the pressure distribution was assumed hydrostatic and the flow was considered quasi-steady in the model. A more detailed description of this case study is provided by Zhang et al.

(2018) and Cohen et al. (2018a). In this paper, the maximum water depth model output along with the 10-m DEM (NED) were used for the FwDET v1.0 and v2.0 calculations.

### 2.3 Applications

Evaluation of FwDET v2.0 operational applications for three case studies is provided:

1.  Hurricane Irene – made landfall along the Mid-Atlantic Coast in late August 2011. To assess flooding from Irene, as part of the CAIR demonstration project, optical remote sensing approaches were used to map water extent, limited to views unobstructed by cloud, high objects like buildings and vegetation. In this study, the highest quality satellite overpass from Landsat was determined to be a Landsat-5 scene obtained on 31 August 2011, five days after the storm made landfall. To identify surface water areas, Landsat-5 surface reflectance was used to compute the modified Normalized Difference Water

Index (mNDWI, Xu 2006). Water detections from the Landsat-5 mNDWI product were then combined with the 2011 National Land Cover Dataset (NLCD) to separate flood areas from known permanent water locations. Areas that were identified as water in the mNDWI product and overlapped with identified water pixels in the NLCD were classified as persistent water. Pixels that were identified as water in mNDWI but not classified as water in the NLCD were determined to be flooded. Due to Landsat-5 imaging occurring after the initial landfall, the scene does not capture the maximum extent of flooding and omits

significant flooding from a storm surge that had receded. A 30-m DEM (NED) was used as input for the FwDET v2.0.

2.  Hurricane Florence – made landfall in North Carolina on September 14, 2018. The slow-moving storm generated large amounts of rainfall over the Carolinas resulting in widespread and severe riverine and pluvial flooding. GFP was activated on September 15. One of the products shared through the GFP network was a daily map of maximum detected flooding at 90-m resolution based on downscaled Advanced Microwave Scanning Radiometer 2 (AMSR2) and GPM Microwave Imager (GMI)

passive microwave satellite observations from the Atmospheric and Environmental Research, Inc. (AER) FloodScan system (https://floodscan.aer.com). Whereas higher resolution remote sensing products were shared by GFP members, the FloodScan maximum flooding product, which included flooding in woody wetlands regions adjacent to observed floodwater, resulted in the most continuous and spatially-extensive flood maps. A 30-m DEM (NED) was used for the FwDET v2.0.

3.  Sri Lanka – torrential monsoon rainfall led to major flooding across Sri Lanka in May 2018. GFP was activated on May 22nd.

As part of the GFP activation, The DFO Flood Observatory (hereafter DFO) published online a flood map based on a 10 m resolution           Sentinel-1b           imagery           (see           DFO           event           page: http://floodobservatory.colorado.edu/Events/4619/2018Somalia4619.html). Several DEM products were tested here as input for the FwDET v2.0 (described later).

## 3 Results and Discussion

### 3.1 FwDET v2.0 Evaluation

Floodwater depth estimates by FwDET v2.0 correspond well with model-simulated water depth for the Norfolk-Portsmouth case study (Figure 2). Maximum floodwater depth (the grid-cell with the highest value) is overestimated by FwDET v2.0 but the water depth rasters yielded similar averages (0.77 m and 0.65 m for the model and FwDET v2.0 respectively) and standard deviation

(0.56 m and 0.58 m for the model and FwDET v2.0 respectively). The mean difference in floodwater depths, calculated by





averaging the raster values of the [FwDET v2.0 – Model] map algebra expression, is -0.16 m with a standard deviation of 0.29 m, meaning that FwDET v2.0 is slightly underestimating floodwater depth. The average absolute difference ([|(FwDET v2.0 – Model)|]) is 0.18 m with a standard deviation of 0.28 m. The histogram distribution (Figure 2e) of the difference map (Figure 2d) shows that the vast majority of grid-cells have a bias of between 0 and -0.33 m and that biases below -1 m and above 0.33m are

rare.

Although the mean difference in water depth estimations by FwDET v2.0 and the hydrodynamics model are small, the difference map (Figure 2d) reveals a heterogenous tapestry of values, some of which are quite considerable and many with sharp (straight line) transitions, not readily apparent in the water depth maps (Figure 2a and b). They are attributed to FwDET v2.0 reliance on nearest boundary cell elevation to calculate water depth. The use of Euclidian distance to assign the nearest boundary

grid-cell can lead to these straight-line transitions as well as inaccuracies in water depth where, for example, backwater effects are driven by complex flow paths. In urban environments, streets and buildings can amplify these biases. In this case study, in which a 1-m LiDAR DEM was used, buildings were identified as having a higher elevation. This created non-flooded areas within the flooded domain which FwDET v2.0 identified as boundary locations. Consequently, some nearby grid-cells erroneously assigned values depicting unrealistic flood boundary elevations. These errors stem from false elevation reflections of sometimes multi-story

building roofs instead of the actual (street level) floodwater elevation. Similarly, other man-made structures (e.g. highways, overpasses, dikes) can lead to these considerable overestimations. Effects of these are visible throughout Figure 2d but are not consistent as some locations around buildings are actually underestimated. Underestimation by FwDET v2.0 is most common near banks and shorelines. This is due to the boundary grid-cells assigned for these locations, which were not from the actual maximum inland extent of floodwater in that location, but of a nearer boundary grid-cell which under-represent the true local floodwater

elevation.

Floodwater depth estimates by FwDET v2.0 also correspond well with model-simulated water depth for the Brazos River case study (Figure 3). Maximum water depths are similar (Figure 3a and b). Average water depth calculated by FwDET v2.0 is 2.1 m, compared to the model's 2.2 m prediction. The standard deviation was also very similar with a value of 2.51 m for FwDET v2.0 and 2.56 m for the model. The difference between the model and FwDET v2.0 water depth calculation is small with an average

of -0.16 m [FwDET v2.0 – Model] and a standard deviation of 0.46 m. Similar to the Norfolk-Portsmouth case study, FwDET v2.0 slightly underestimates floodwater depth. The absolute difference in water depth is 0.31 m with a standard deviation of 0.46 m. The histogram distribution (Figure 3e) of the difference map (Figure 3c) shows that the vast majority of grid-cells have a bias of between -0.33 and 0.33 m with a small proportion of grid-cells having a bias of over 1 m and below-1.33 m.

The difference map (Figure 3c) shows that the largest biases in FwDET v2.0 are mostly concentrated along the river

channel. These are likely due to the hydraulic slope, which is simulated by the hydrodynamics model but are not expressed in FwDET topography-based approach. As these biases relate to the active river channel, their implications for flood applications are small. Other regions of relatively high biases are along the western edges of the flooded domain. These are artifacts of the model domain set up. As described in Cohen et al. (2018a) and Zhang et al. (2018), the iRIC simulation grid extent and resolution must be manually defined by the user affecting the fluid dynamics along the edges of the simulation domain.

FwDET v2.0 shows a slight improvement over v1.0, which had a mean water depth of 1.95 m and an absolute difference of 0.37 m. A comparison in the two versions bias against the model results (Figure 3d), calculated as [|FwDET v2.0 - Model| – |FwDET v1.0 - Model|], has an average value of 0.01 m and standard deviation of 0.33 m. The differences between v1.0 and v2.0 are expected to be small for riverine flooding, as the elevation of all boundary grid-cells is higher than zero. The small improvement in FwDET v2.0 is primarily due to the use of a different tool for assigning the nearest boundary elevation ('Focal Statistics' loop

in v1.0 vs. 'Cost Allocation' in v2.0).


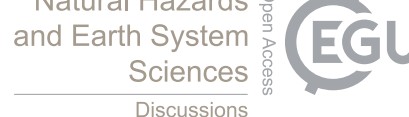

The Brazos River case study was also used to compare the runtime improvement of FwDET v2.0. At 10 m resolution the domain has 2,087x1,816 (3,789,992) grid-cells requiring 100 iterations of the 'Focal Statistics' loop in FwDET v1.0. FwDET v1.0, v2.0, and QGIS versions were run on a Windows 7 desktop with two Intel Xenon E5 2670 2.5 GHz processors and 64GBs of RAM. FwDET v1.0 runtime was 24 minutes and 14 seconds, FwDET v2.0 runtime was 1 minute 33 seconds, and FwDET QGIS

runtime was 49 seconds. Yielding a runtime improvement of over a factor of 15 between v1.0 and v2.0 and a further improvement by a factor of nearly 2 between v2.0 and QGIS. Faster runtime by FwDET QGIS is due to its use of GDAL's raster clipping tool which is written in C (GDAL, 2019) and, similar to v2.0, there is no iterative loop. This clipping procedure (used in all FwDET versions) ensures that floodwater depth is rendered only within the flooded domain (see Cohen et al., 2018a). FwDET v2.0 ArcGIS script tool allows users to provide a pre-clipped DEM to reduce runtime. This is mostly useful when repeated runs for the same

inundation extent are conducted.

### 3.2 FwDET v2.0 Application Results

Large-scale coastal flooding, typically associated with tropical cyclones, is challenging to analyze from both an observational (remote sensing, point data) and modeling perspectives. This is because of the diversity in land cover and flooding sources. Storm

surge, for example, can be highly energetic but short in duration relative to riverine flooding. That could create observational challenges for remote sensing applications. The NASA CAIR project (Rogers et al., 2018, NASA CAIR, 2018) utilized FwDET v2.0 to demonstrate the ability to integrate satellite-derived earth observations and physical models into actionable knowledge. The integration of observations and models allow for a more comprehensive understanding of the compounding risk experienced in coastal regions. The demonstration produced flood inundation maps to predict building-level impacts of a representative storm

in the mid-Atlantic region for Hurricane Irene. FwDET v2.0 used best-available remotely sensed imagery to determine inundation depth immediately following the storm.

To estimate floodwater depth following Irene, the Landsat 5 floodwater classification was converted to a polygon layer (Figure 4) to be used as input for FwDET v2.0. A DEM for the region was compiled by mosaicking the corresponding 30 m spatial resolution NED tiles. While 10 m NED products are available for this region, the 30 m product was used given the resolution of

the flood inundation source (30 m Landsat 5 images) and the size of the domain (Figure 4). Although there are insufficient ground-based observations to make a quantitative accuracy determination, overall the spatial trends in water depth estimation seem reasonable. The average floodwater depth for the entire domain was 0.64 m with a maximum of 41.7 m. The latter is obviously an over-prediction resulting from the misclassification of floodwater from the satellite imagery or spatial mismatch between the inundation map and the DEM. Zooming in on the Norfolk-Portsmouth area reveals a much smaller flooding extent compared to

the model simulation results (Figure 2). This is reasonable given that the model simulated the maximum flooding conditions during the event while the flood inundation layer used here is based on a Landsat 5 image captured 5 days past the Hurricane's landfall. Under-prediction of flood extent may also be due to challenges in floodwater classification in urban environments at this resolution. Average floodwater depth calculated by FwDET v2.0 for the Norfolk-Portsmouth domain was 0.85 m, which is slightly higher than water depth predictions for the same domain in the high-resolution maximum extent case study (Figure 2; 0.77 m and 0.65 m

for the model and FwDET v2.0 respectfully). The likely cause for this overestimation in this coastal urban region is the resolution of the inundation map and DEM relative to the area's small topographic gradient and fragmentation of the flooded domain. Thus, the use of FwDET v2.0 over large coastal flood domains can be useful for providing a synoptic overview of flood severity, but its localized analysis, especially for urban areas, should be considered in the context of the input data used (mainly its resolution).

For the Hurricane Florence application case study, FwDET v2.0 estimates (Figure 5) show high water depths (over 3 m)

within the major river floodplains and shallow to medium water depths (below 3m) across the flooded domain. Average floodwater



depth is 0.92 m with a standard deviation of 1.7m. This is a reasonable result given the other case studies (we do not have comprehensive observed or simulated water depth data for a quantitative assessment). Maximum estimated water depth is 39.6 m which is clearly an overestimation, even though calculations include permanent water features. That is because DEMs typically capture the water surface elevation of permanent water features (see Figure 1).

5        For the Sri Lanka application case study, the flood inundation map produced by DFO was highly fragmented in most parts of the country leading to many small inundation polygons (Figure 6). As described earlier, fragments in the inundation extent, assuming it represents reality, can be advantageous as it can shorten the distance to the nearest boundary grid-cells which may yield more accurate (localized) water elevation estimation by FwDET v2.0. However, a highly fragmented inundation extent can be problematic if the flooded sections are small relative to the input DEM resolution and the local terrain gradient. For example, a

flooded area with an extent of only a few DEM grid-cells in a flat area may result in a negligible water depth because the elevation of the boundary and inundated grid-cells are similar. High-resolution DEMs outside the U.S. are often difficult to obtain as most countries do not openly share national DEMs, or they do not exist. As a result, in emergency response situations, we often can only use global DEMs as input to FwDET (see Cohen et al., 2018a, 2018b). For this event three DEM products were tested:

1.  HydroSHEDS (based on the Shuttle Radar Topography Mission (SRTM) DEM); 3 arc-sec (~90m) resolution.

2.  Multi-Error Improved–Terrain (MERIT; Yamazaki et al., 2017); 3 arc-sec (~90m) resolution; http://hydro.iis.u-tokyo.ac.jp/~yamadai/MERIT_DEM/index.html

3.  ALOS; 30m resolution; http://www.eorc.jaxa.jp/ALOS/en/aw3d30/index.htm

         While ALOS offers the highest spatial resolution, it is distributed as an integer raster which means that its vertical resolution is effectively 1 m. This is a considerable disadvantage, especially in low slope terrains. MERIT resulted in improved

depth estimations over using HydroSHEDS (not shown here) but its horizontal resolution is inadequate considering the resolution of the remote sensing inundation map (10 m) and the high degree of fragmentation in the inundation extent input. Use of the ALOS DEM yielded the most appropriate floodwater depth map (Figure 6) for the Sri Lanka flood, but with a relatively high degree of uncertainty due to its limited vertical resolution, a high degree of fragmentation relative to the DEM vertical resolution, and mismatch in horizontal resolution between the inundation map and DEM. This event demonstrated challenges associated with

high-resolution DEM availability. This case study highlights the need to carefully consider the appropriateness of DEM choice in the context of the resolution and nature of the inundation extent map.

## 4 Conclusions

The Floodwater Depth Estimation Tool (FwDET) calculates water depth based solely on an inundation polygon and a DEM. This

enables rapid application over large domains and globally, which is highly advantageous for disaster response and large scale or frequent (many flood map) uses. The first version (v1.0) of FwDET was used extensively in the last two years as part of flood response activations by the Global Flood Partnership. A new version of FwDET is presented here. FwDET v2.0 enables floodwater depth calculation in coastal areas (while maintaining its riverine capabilities) by modifying the flood boundary identification approach and improving runtime. The latter is important given the need for hyper-fine resolution DEMs to represent low slope

coastal topography.

         FwDET v2.0 calculation accuracy was estimated by comparing its water depth calculations against those from physically-based hydrodynamic simulations. Two case studies were used here for evaluation: coastal flooding in Norfolk-Portsmouth during the 2011 Hurricane Irene, using 1 m LiDAR DEM, and riverine flooding in Brazos River (TX) in 2016, using 10 m DEM. In both cases FwDET v2.0 corresponded well with the hydrodynamic simulations, yielding an average difference of 0.18 m and 0.31 m

for the Norfolk-Portsmouth and Brazos case studies respectively. The average and standard deviation of the two water depth





products (FwDET and model-simulated) were also similar. FwDET v2.0 considerably over-predicted maximum flood depth (a grid-cell with the highest value). This can be due to mismatches between the flood boundary and the DEM or inaccurate identification of the appropriate flood boundary grid-cell. In FwDET the nearest boundary grid-cell for each grid-cell within the flooded domain is identified based on Euclidian distance. However complex fluid dynamics and flow paths can result in local

floodwater elevation which differs from the nearest boundary grid-cell. These errors are due to the simplicity of FwDET and can lead to unrealistic water depth patters in some locations. The results from this and past papers demonstrate that FwDET can be considered a first-order tool for providing a synoptic overview of floodwater depth distribution. Its ability to provide estimates at finer scales depends on the spatial complexity of the flooded domain and the resolution of the flood extent map and DEM. Generally, simple flood extents and good correspondence between the inundation map and DEM will yield more accurate depth

estimations.

FwDET v2.0 was compared to v1.0 using the Brazos case study. Results show that, as expected, the two versions yielded very similar water depth maps for this riverine case study. FwDET 2.0 was able to achieve a considerable improvement in runtime (by a factor of 15) an additional improvement (by a factor of 2) with its QGIS version. The QGIS version does not yet include all the methodological improvements of FwDET v2.0 and should not be applied for coastal flood analysis. FwDET v1.0 and v2.0 are

freely available as Python scripts and ArcGIS and QGIS tools through the Community Surface Dynamics Modeling System (CSDMS) Model Repository (https://csdms.colorado.edu/wiki/Model:FwDET) and the Surface Dynamics Modeling Lab (SDML) website (https://sdml.ua.edu/models/).

A FwDET application using remotely sensed flood maps to estimate flood depth was demonstrated for large scale flooding during the 2011 Hurricane Irene on the mid-Atlantic U.S. coast (30 m Landsat multi-spectral images), the 2018 Hurricane Florence

that affected North and South Carolina (90 m downscaled AMSR2 and GMI passive microwave images), and the 2018 flooding in Sri Lanka (10 m Sentinel-1 SAR image). These case studies demonstrated the functionality of FwDET v2.0 in providing a synoptic assessment of floodwater depth. Limitations in using FwDET were highlighted, including challenges in obtaining high spatial resolution DEMs and increases in uncertainty when applied to highly fragmented flood inundation extents. The accuracy and timing, relative to the flood peak, of the remotely sensed flood map, is likely to be the greatest source of uncertainty in FwDET

flood depth estimations.

**Acknowledgments**

This project was funded as part of the NASA Applied Sciences Disasters Program Mid-Atlantic Communities and Areas at Intensive Risk (CAIR) demonstration project.

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

**Figures and captions**

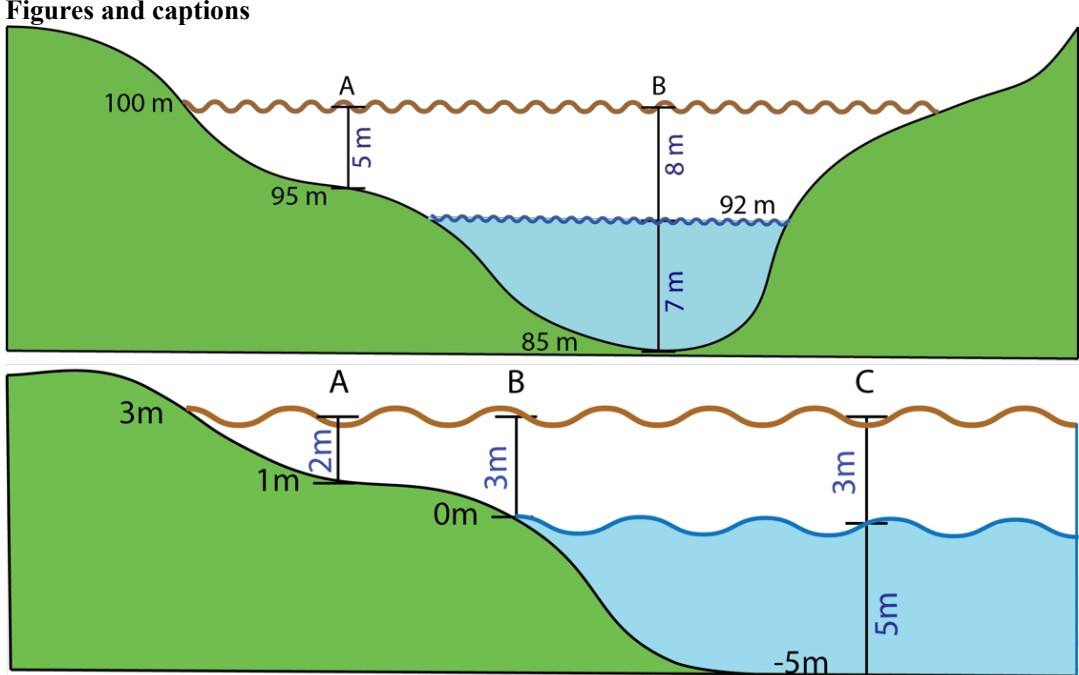

**Figure 1. Theoretical floodplain (top panel) and coastal (bottom panel) cross sections illustrating FwDET floodwater depth calculation
approach. The elevation (amsl; black numbers) of the floodwater boundary (100m at the top and 3m in the bottom) are used to calculate
10   water depth (blue numbers) for each grid-cell within the flooded domain (point A). In riverine flooding (top panel) underestimation of
water depth is expected over the river (point B) as DEMs typically capture the water surface elevation. In coastal flooding (bottom panel)
the seaward flood boundary can be at the coastline (point B) or over the ocean (point C) and cannot be used to estimate floodwater depth
(elevation ≤ 0). In FwDET v2.0 these boundary locations are excluded which means that only the inland flood boundary is used.**





**Figure 2. The Norfolk-Portsmouth August 2011 Hurricane Irene flooding case study: (a) simulation domain, location overview map (bottom right inset), and the zoom-in extent used in panels (b-d) over the Lafayette River tidal estuary (red box); (b) model-simulated maximum water depth map; (c) FwDET v2.0 floodwater map; (d) difference map between (b) and (c) [FwDET v2.0 – Model]; (e) histogram of grid-cells in the difference map (d). Background sources: Esri, DigitalGlobe, GeoEye, i-cubed, USDA FSA, USGS, AEX, Getmapping, Aerogrid, IGN, IGP, swisstopo, and the GIS User Community**





**Figure 3. The May 2016 flood event for the Brazos River (Texas) case study: (a) Model-simulated maximum water depth; (b) FwDET v2.0 estimated floodwater depth; (c) difference map between (a) and (b) [FwDET v2.0 – Model]; (d) comparison between FwDET v1.0 and v2.0 estimation accuracy [|v2.0 - Model| – |v1.0 - - Model|] (positive values indicate smaller bias by FwDET v2.0); (e) histogram of grid-cell values of the difference map (c). Background sources: Esri, DigitalGlobe, GeoEye, i-cubed, USDA FSA, USGS, AEX, Getmapping, Aerogrid, IGN, IGP, swisstopo, and the GIS User Community.**





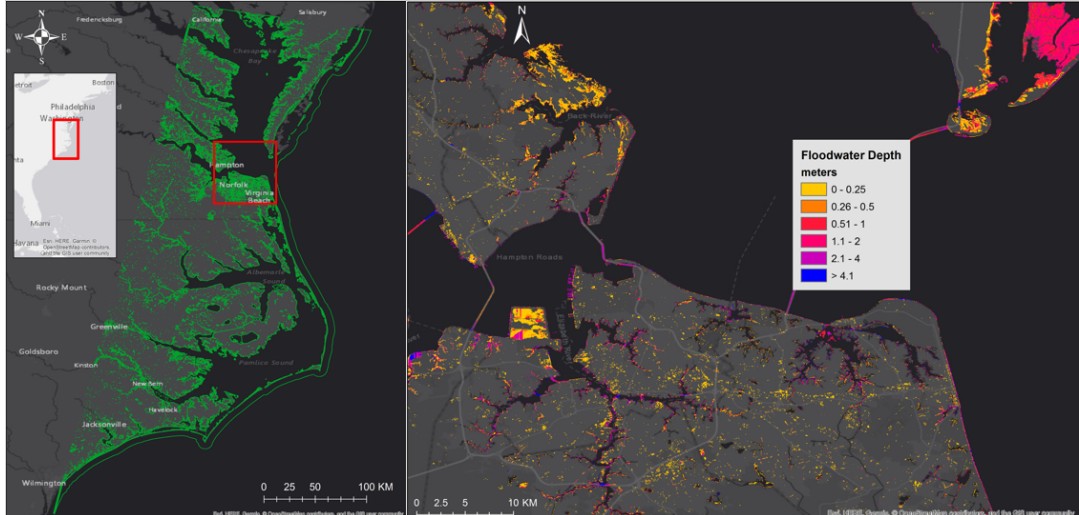

**Figure 4. Flooded domain and overview map (left panel) of the 2011 flooding following Hurricane Irene landfall in the U.S. Mid-Atlantic coast. The flood inundation was classified from two Landsat TM images which were used as input in FwDET v2.0 to calculate**
5  **floodwater depth (right panel zooming-in on the Norfolk-Portsmouth area). Background Sources: Esri, DeLorme, HERE, MapmyIndia.**

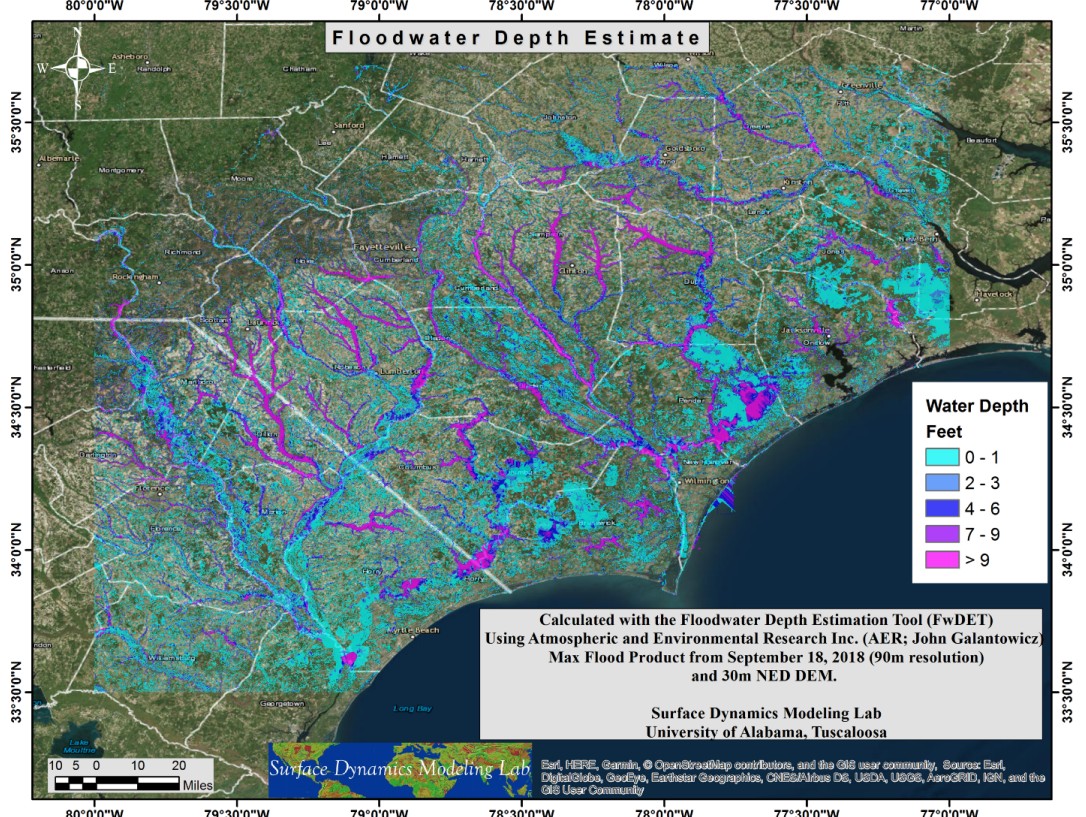

**Figure 5. A floodwater depth map which was shared during the GFP Hurricane Florence activation. Water depth was calculated with**
10  **FwDET v2.0 using a maximum inundation extent map (as of September 18, 2018) and a 30 m DEM (NED). Units were converted to**





**English scales for better integration in U.S. emergency response entities. Background sources: Esri, DigitalGlobe, GeoEye, i-cubed, USDA FSA, USGS, AEX, Getmapping, Aerogrid, IGN, IGP, swisstopo, and the GIS User Community.**

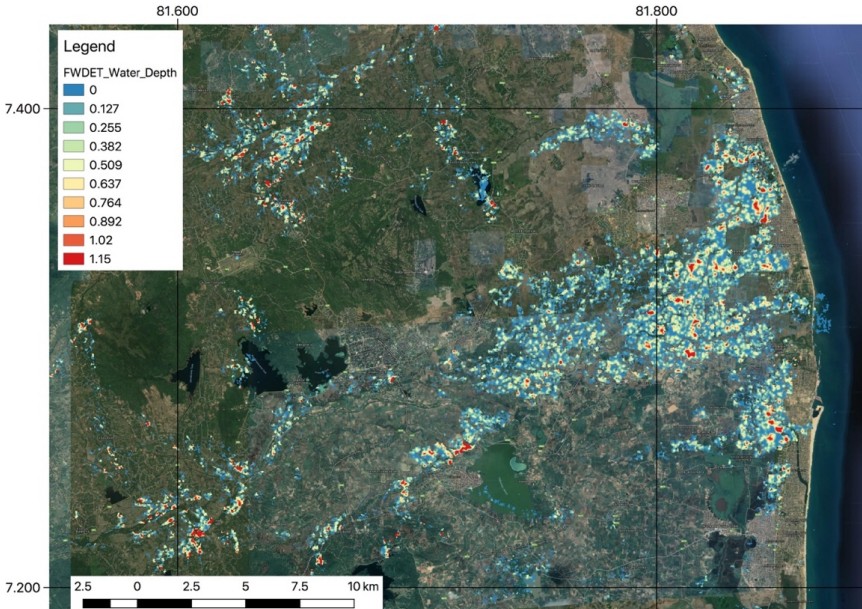

5  **Figure 6. A floodwater depth map shared during the GFP activation for the 2018 flooding in Sri Lanka, zooming in on the south-east part of the country. Water depth was calculated using FwDET v2.0 with a 10m flood inundation map (for May 19, 2018) and 30m DEM (ALOS). Background sources: Esri, DigitalGlobe, GeoEye, i-cubed, USDA FSA, USGS, AEX, Getmapping, Aerogrid, IGN, IGP, swisstopo, and the GIS User Community.**

