# Peer review of "The Floodwater Depth Estimation Tool (FwDET v2.0) for Improved Remote Sensing Analysis of Coastal Flooding"

_Natural Hazards and Earth System Sciences, 2019_

## Referee Comment (RC1) · Anonymous Referee #1 · 6 May 2019

This article presents a clever approach to estimate flood depths using high resolution flood extent images from remote sensing, and digital elevation maps. Given the widespread availability of remote sensing products, the main challenge of such type of products is finding good quality DEM in all world regions, as also mentioned by the authors. The operational use of the FwDET tool in near real time during flood emergencies gives it additional importance, and stresses the need for a computationally efficient tool as demonstrated in this article, in comparison to the previous version. The article is relatively short, though self-contained, and is rather convincing about the added value of version 2.0 of the tool. I'd appreciate a clearer description of methods, at times a bit unclear. Also, evaluation based on actual measurements would give

much more strength to the model (see further comments below). Last point, I think that the choice of the modeled inundation map to compare with the FwDET estimate can change substantially the results. It would be useful to see a sensitivity analysis of choosing different time steps around the peak of simulated inundation maps, to understand the limitations of choosing the maximum flood depths, as currently done (more comments below).

Specific comments P2 l27: It was also recently used [. . .] P4 l24: is 3 a footnote or part of the model name (UnTRIM)? P4 l35: please add some more quantitative details to the statement "compared favorably" P4 l37-38: are the authors assuming that the remote sensing floodwater extent is representative of the maximum extent for this event? This assumption should be better clarified and its soundness proved. Why not simply using the simulated map which is the closest in time to the moment when the flood image was taken? The same considerations apply to case #3 (p5 l5-6) P6 l7-8: these sharp transitions are not really visible with the current zoom level. Sect 3: the authors could comment on the availability or not of point measurements. Including these in the comparison would strengthen the results. This can also be inferred from photos taken during the flooding near known features, such as buildings, bridges etc. Figure 4, right panel: The current legend location on top of the Chesapeake Bay Bridge is unusual and gives the impression to the reader that there is something to hide. Please move the legend on a sea area. Figure 6: Units are missing in the legend

---

## Referee Comment (RC2) · Anonymous Referee #2 · 29 Jul 2019

This paper documents an incremental improvement to the FwDET tool making it more suitable for flood depth estimation along coastal or permanent water body locations. Additionally, the paper documents substantial improvement in the technical aspects of the tool by converting it to Python and making it more open for community use. While the tool is good the presentation of it in the paper could use some improvement to show the true value of the 2.0 version of FwDET.

Specific comments:

Pg 2, line 13: This sentence is unsupported here and hard to believe given the uncertainty present in the depth estimates. Is a depth estimate with uncertainty +/- 0.33 m

more useful to rescue and relief efforts than a map of low water crossings? How will this information be used by decision makers in real time?

Pg. 4 line 2: It may be helpful to talk about how much work is necessary to prepare the cost raster from the land-cover map. Is this a quick process, or will these need to be precomputed for real use?

Pg. 5 line 8: Is the QGIS version of the script really FwDET 2.0 or is it FwDET 1.0? Either way this is an excellent improvement, but clarification on the version may be useful to the readers.

Pg. 6 line 2: It would be helpful to express the error as a percentage of the overall depth. A error of 0.18m sounds small, but it is an error of about 50% of the observed heights. For the Brazos River, the depths quadruple to about 2 m, but the error stays constant at around 0.16 m. So this method performs much better for deeper water situations, or there is an inherit limitation to the method that results in a lower bound on the error of around 0.15 m?

Pg. 7 line 5: Very impressive performance speed up!

Pg. 8 line 5: Is the fragmentation due to cloud cover? If so should future work proposed be how to extrapolate with FwDET to regions between fragments?

Overall: While the Brazos River example is compared for both FwDET 1.0 and FwDET 2.0, there is no example illustrating the problem at coast lines for FwDET 1.0 and how FwDET 2.0 solved the problem. The description of the improvements to the method could also benefit from better clarity on how locations are being chosen. For example, the line artifacts are attributed to this too so it may be useful to have a map showing just the locations used for the depth estimation with the FwDET 1.0 vs 2.0. Another example is a float-integer-float trick is mentioned, but not described what it is or how it is used. The methodology is also missing a description of how the modeled inundation rasters were converted to polygons for FwDET? Simply water depth > 0.0, or is there

smoothing applied or another threshold chosen?
* * *

---

## Author Comment (AC1) · 31 Jul 2019

This article presents a clever approach to estimate flood depths using high resolution flood extent images from remote sensing, and digital elevation maps. Given the widespread availability of remote sensing products, the main challenge of such type of products is finding good quality DEM in all world regions, as also mentioned by the authors. The operational use of the FwDET tool in near real time during flood emergencies gives it additional importance, and stresses the need for a computationally efficient tool as demonstrated in this article, in comparison to the previous version. The article is relatively short, though self-contained, and is rather convincing about the added value

of version 2.0 of the tool. – We thank the referee for the time and effort in reviewing the manuscript. The comments are useful and constructive. See our response to each point below. Revised manuscript with Track Changes is attached as 'Supplement'

I'd appreciate a clearer description of methods, at times a bit unclear. –The methodology was clarified in response to Referee #2 comments and our own new review.

Also, evaluation based on actual measurements would give much more strength to the model (see further comments below). – See our response to these comments below.

Last point, I think that the choice of the modeled inundation map to compare with the FwDET estimate can change substantially the results. It would be useful to see a sensitivity analysis of choosing different time steps around the peak of simulated inundation maps, to understand the limitations of choosing the maximum flood depths, as currently done (more comments below). – See our response to these comments below.

Specific comments: P2 l27: It was also recently used [. . .] – changed

P4 l24: is 3 a footnote or part of the model name (UnTRIM)? –This is how it was referenced in the original paper, but we now changed it in the text to 'Version 3' for clarification.

P4 l35: please add some more quantitative details to the statement "compared favorably" —The sentence was changed to: "The street-level model subsequently generated hourly inundation outputs for floodwater depths in meters, which compared favorably with three municipally-owned water level sensors (RMSE = 4.61 cm), and 18 USGS-reported high water marks (RMSE = 9.73 cm) in southeast Virginia, as noted in Loftis et al. (2018, 2019)."

P4 l37-38: are the authors assuming that the remote sensing floodwater extent is representative of the maximum extent for this event? This assumption should be better clarified and its soundness proved. Why not simply using the simulated map which is the closest in time to the moment when the flood image was taken? The same considerations apply to case #3 (p5 l5-6). –The FwDET calculation is not compared to remote sensing observations in these two case studies as explained at the start of this section (p4 l15). This is because the tool only calculates water depth, and thus in order to evaluate it we need an independent source which specifically provides water depth. To accomplish this, and to isolate the tool's bias due to its calculation approach, we use the modeled flood extent rather than a remote sensing derived one. Potential biases due to the remote sensing input are looked at with the other case studies.

P6 l7-8: these sharp transitions are not really visible with the current zoom level. – We added a red circle in figure 2d to direct the viewer (in the text) to an example of these artifacts.

Sect 3: the authors could comment on the availability or not of point measurements. Including these in the comparison would strengthen the results. This can also be inferred from photos taken during the flooding near known features, such as buildings, bridges etc. –There were very few high-water mark observations for case study #1 and none for case study #2. In response to the reviewer's previous comment, we added more information about the model accuracy. That being said, and as explained in the methodology, the two model case studies are used here as true water depth results in order to isolate biases in the methodology due to inundation extent observation (i.e. remote sensing). In both cases the models were calibrated against observations.

Figure 4, right panel: The current legend location on top of the Chesapeake Bay Bridge is unusual and gives the impression to the reader that there is something to hide. Please move the legend on a sea area. –Done. The map extent was also changed to align it with Figure 2.

Figure 6: Units are missing in the legend. –Added

Please also note the supplement to this comment:

https://www.nat-hazards-earth-syst-sci-discuss.net/nhess-2019-78/nhess-2019-78-AC1-supplement.pdf

**Supplement:**

[revised manuscript text omitted]

---

## Author Comment (AC2) · 31 Jul 2019

This paper documents an incremental improvement to the FwDET tool making it more suitable for flood depth estimation along coastal or permanent water body locations. Additionally, the paper documents substantial improvement in the technical aspects of the tool by converting it to Python and making it more open for community use. While the tool is good the presentation of it in the paper could use some improvement to show the true value of the 2.0 version of FwDET. – We thank the referee for the time and effort in reviewing the manuscript. The comments are useful and constructive. See our response to each point below.

Specific comments: Pg 2, line 13: This sentence is unsupported here and hard to believe given the uncertainty present in the depth estimates. Is a depth estimate with uncertainty +/- 0.33 m more useful to rescue and relief efforts than a map of low water crossings? How will this information be used by decision makers in real time? – This work was actually motivated by end-user requests from the Dartmouth Flood observatory and personal discussion with first responders (particularly the former chief of staff of the Austin Fire Department). For first responders, water depth information is valuable for assessing road accessibility and getting a sense of danger to people and vehicles. The sentence is generic and does not refer to a specific tool or methodology; it is meant to describe the key motivation for this study by emphasizing the importance of water depth information. There is also no claim that water depth estimation is necessarily more important than other information.

Pg. 4 line 2: It may be helpful to talk about how much work is necessary to prepare the cost raster from the land-cover map. Is this a quick process, or will these need to be precomputed for real use? – This was added to the sentence: "... (through e.g. identification of permanent water bodies) ..."

Pg. 5 [4] line 8: Is the QGIS version of the script really FwDET 2.0 or is it FwDET 1.0? Either way this is an excellent improvement, but clarification on the version may be useful to the readers. – Good point. We added this sentence: "It is therefore not a full solution of FwDET v2.0 but it does include the coastline boundary cell identification procedure."

Pg. 6 line 2: It would be helpful to express the error as a percentage of the overall depth. A error of 0.18m sounds small, but it is an error of about 50% of the observed heights. –These values were added to both case studies (24% and 14% of the model mean average depth)

For the Brazos River, the depths quadruple to about 2 m, but the error stays constant at around 0.16 m. So this method performs much better for deeper water situations, or there is an inherit limitation to the method that results in a lower bound on the error of around 0.15 m? – No, the error goes up to 0.31m but it is true that the relative error is reduced from 24% to 14%. This is likely because the river itself is included in the statistics which is deep and 'easy' for the tool to calculate. This point was added in the text (section 3.1): "The lower relative bias in the Brazos case study compared to the Norfolk-Portsmouth case study is likely due to the inclusion of the river itself in the statistical calculations. The river segment is relatively deep, and its water depth is relatively easy to estimate (not considering its true bathymetry)."

Pg. 7 line 5: Very impressive performance speed up! –Thank you!

Pg. 8 line 5: Is the fragmentation due to cloud cover? If so should future work proposed be how to extrapolate with FwDET to regions between fragments? – No, this seems to be a more or less accurate classification given the sensor resolution. We added this sentence: "The remote sensing classification used appears to be accurate representation of ground conditions given the sensor resolution."

Overall: While the Brazos River example is compared for both FwDET 1.0 and FwDET 2.0, there is no example illustrating the problem at coast lines for FwDET 1.0 and how FwDET 2.0 solved the problem. The description of the improvements to the method could also benefit from better clarity on how locations are being chosen. For example, the line artifacts are attributed to this too so it may be useful to have a map showing just the locations used for the depth estimation with the FwDET 1.0 vs 2.0. – FwDET 1.0 does not work at coastal regions as the boundary elevation at the coastline is lower than the flooded domain. As a result, all the cells closest to the coastline (relative to the inland boundary) receive a no-data value. There is therefore no point in comparing the two versions in these locations. This was now clarified in the text: "A Comparison to FwDET 1.0 is not valuable for this (or any coastal) case study. This is because FwDET 1.0 does not work at coastal regions as the boundary elevation at the coastline is lower than the flooded domain. As a result, all the cells closest to the coastline (relative to the inland boundary) receive a no-data value."

Another example is a float-integer-float trick is mentioned, but not described what it is or how it is used. – We added this to the relevant sentence: ". . . (multiplication of the DEM by 106 and then dividing it by the same factor after the tool run) . . ."

The methodology is also missing a description of how the modeled inundation rasters were converted to polygons for FwDET? Simply water depth > 0.0, or is there smoothing applied or another threshold chosen? – We added this sentence in section 2.2: "The rasters are converted to FwDET inundation extent input polygons by re-classifying all non-zero water depth cells as 1 and using the ArcGIS (or equivalent QGIS) 'Raster to Polygon' tool to generate a feature layer."

Please also note the supplement to this comment:
https://www.nat-hazards-earth-syst-sci-discuss.net/nhess-2019-78/nhess-2019-78-AC2-supplement.pdf

**Supplement:**

[revised manuscript text omitted]

---

## Author Response (AR1)

**Responses to correction listed by the Associate Editor**

*Thanks for your revisions in response to the reviewers' comments. I think the manuscript is in good shape, and I'm prepared to accept it subject to a few very minor grammatical/typographical corrections, which are listed separately.*

Thank you.

Non-public comments to the Author:

*page 3, lines 3-4: do you mean "deducing", as in "figuring out"? Or do you mean "deducting", as in "subtracting" (but in that case, it is presumably the ground height that is subtracted from the water height, not the other way around). Same issue on line 12.* -- We substituted "deducting" with "subtracting". The order of subtraction [floodwater – topographic] is correct as it is floodwater elevation (in amsl) which is higher than the topographic elevation.

*page 3, line 16, insert "do" after "often"* – Done

*page 3, line 34 "assign" (singular)* -- Done

*page 4, line 16; page 5, lines 2-3: delete comma after "al."* – Done

*page 7, lines 17-18: incomplete sentence (try "The new algorithm yields ..." or something like that)* -- Fixed

*page 7, line 26: "perspective" (singular)* -- Done

*page 8, line 23: "elevations" (plural)* -- Done